# A Difficult Case of Ventriculitis in a 40-Year-Old Woman with Acute Myeloid Leukemia

**DOI:** 10.3390/antibiotics13050432

**Published:** 2024-05-10

**Authors:** Raffaella Rubino, Marcello Trizzino, Luca Pipitò, Giuseppe Sucato, Marco Santoro, Rosario Maugeri, Domenico Gerardo Iacopino, Giovanni Maurizio Giammanco, Sergio Siragusa, Antonio Cascio

**Affiliations:** 1Infectious and Tropical Diseases Unit, Sicilian Regional Reference Center for the Fight against AIDS, AOU Policlinico “P. Giaccone”, 90127 Palermo, Italy; marcello.trizzino@policlinico.pa.it (M.T.); antonio.cascio03@unipa.it (A.C.); 2Antimicrobial Stewardship Team, AOU Policlinico “P. Giaccone”, 90127 Palermo, Italy; 3Infectious and Tropical Diseases Unit, Department of Health Promotion, Mother and Child Care, Internal Medicine and Medical Specialties “G. D’Alessandro”, University of Palermo, 90127 Palermo, Italy; lucapipito@gmail.com; 4Hematology Unit, University Hospital Paolo Giaccone, 90127 Palermo, Italy; giuseppe.sucato@policlinico.pa.it (G.S.); marco.santoro@policlinico.pa.it (M.S.); sergio.siragusa@policlinico.pa.it (S.S.); 5Neurosurgical Clinic, AOU Policlinico “P. Giaccone”, Post Graduate Residency Program in Neurologic Surgery, Department of Biomedicine Neurosciences and Advanced Diagnostics, School of Medicine, University of Palermo, 90127 Palermo, Italy; rosario.maugeri1977@gmail.com (R.M.); gerardo.iacopino@gmail.com (D.G.I.); 6UOC of Microbiology, Virology and Parasitology, Department of Health Promotion, Mother and Child Care, Internal Medicine and Medical Specialties “G. D’Alessandro”, University of Palermo, 90127 Palermo, Italy; giovanni.giammanco@unipa.it; 7Department of Health Promotion, Mother and Child Care, Internal Medicine and Medical Specialties “G. D’Alessandro”, University of Palermo, 90127 Palermo, Italy

**Keywords:** ventriculitis, multidrug resistance, meropenem/vaborbactam, central nervous system infection

## Abstract

Ventriculitis and nosocomial meningitis caused by carbapenem-resistant Gram-negative and vancomycin-resistant Gram-positive bacteria represent a growing treatment challenge. A case of ventriculitis and bacteremia caused by carbapenem-resistant, KPC-producing *Klebsiella pneumoniae* and vancomycin-resistant *Enterococcus faecium* in a young woman with acute leukemia who was successfully treated with meropenem/vaborbactam (MVB), rifampicin, and linezolid is described in this paper. This case report emphasizes the importance of a multidisciplinary strategy, including infectious focus control, for the treatment of device-associated central nervous system (CNS) infections from multidrug-resistant bacteria. Considering the novel resistance patterns, more research on drug penetration into the central nervous system, as well as on the necessity of association therapies, is needed.

## 1. Introduction

The bacteria that cause ventriculitis and nosocomial meningitis are typically different from those that cause community-acquired meningitis, including multidrug-resistant (MDR) Gram-positive and Gram-negative bacteria, and various pathogenic pathways are linked to the diseases’ development [1]. The incidence of ventricular catheter-related ventriculitis ranges between 0 and 45%, while that of external ventricular drain (EVD)-related ventriculitis ranges between 0 and 22% [2,3,4]. These infections may be difficult to identify because the changes in the cerebrospinal fluid (CSF) parameters are typically subtle, making it difficult to tell whether the anomalies are caused by infection or device placement [1].

Moreover, antimicrobial resistance is linked to increased morbidity and death [5]. *Klebsiella pneumoniae* carbapenemase (KPC) is the most prevalent carbapenemase in carbapenem-resistant Enterobacterales in Italy [6]. The most recent treatment guidelines for carbapenem-resistant Gram-negative bacteria recommend ceftazidime/avibactam (CAZ/AVI) and meropenem/vaborbactam (MVB)—and, as a second choice, imipenem/relebactam and cefiderocol—as effective antibiotics against carbapenem-resistant, KPC-producing Enterobacterales [7,8,9]. Unfortunately, there are little data in the literature to support the use of these new drugs in CNS infections [10,11,12,13,14,15,16]. Vancomycin-resistant enterococci (VRE) are becoming increasingly prevalent nosocomial pathogens all over the world, but they seldom cause CNS infections. Because of the drugs’ CNS penetration and changing resistance patterns, meningitis and ventriculitis caused by vancomycin-resistant enterococci (VRE) are difficult to treat [17]. In this report, a case of ventriculitis and bacteremia caused by carbapenem-resistant, KPC-producing *K. pneumoniae* and vancomycin-resistant *Enterococcus faecium* in a young woman with acute leukemia who was successfully treated with meropenem/vaborbactam, rifampicin, and linezolid is described.

## 2. Case Report

A 40-year-old woman with an unremarkable family and personal medical history presented at our hospital’s emergency department in October 2022 with severe asthenia and fever resistant to the paracetamol and antibiotics provided by the attending physician for around 15 days. She had been treated with amoxicillin/clavulanate initially, and subsequently, with levofloxacin. At admission, she was pale, prostrate, feverish, and had splenomegaly on physical examination. A blood count revealed normal white blood cells (WBC 5.580/mm^3^) with neutropenia (520/mm^3^, 8.9%), lymphocytosis (2.900/mm^3^, 50.2%), and monocytosis (2.350/mm^3^, 40.7%), as well as moderate anemia and severe thrombocytopenia (15,000/mm^3^). Her C-reactive protein (CRP) level was 90.2 mg/L. Here blood cultures were negative. An ultrasound of the abdomen confirmed the splenomegaly, and a lung computed tomography (CT) scan excluded pneumonia. A bone marrow aspiration and biopsy confirmed the clinical suspicion of acute myeloid leukemia. After a seven-day cycle of piperacillin/tazobactam and steroids with defervescence and a marked reduction in the inflammatory indices, the patient was enrolled in the AML 1819 protocol and received daunorubicin, cytarabine, and gemtuzumab ozogamicin-induction chemotherapy. Here feverish neutropenia (WBC of 780/mm^3^ and neutrophils of 90/mm^3^ (11.6%) on the second day post-induction; WBC of 380/mm^3^ and neutrophils of 0/mm^3^ on the thirteenth day post-induction) worsened in the post-induction period, and treatment with meropenem and levofloxacin was initiated. Blood cultures collected during the fever were positive for *Enterococcus faecium* and, based on the antibiogram, levofloxacin was substituted with daptomycin, achieving initial defervescence and clinical improvement. Because of the onset of severe headache, the patient underwent a brain CT scan, which revealed subarachnoid hemorrhage. Due to the progressive decline in the neurological conditions at the onset of a comatose state, with evidence of obstructive hydrocephalus on a controlled brain CT, a ventricular drain was positioned. During meropenem and daptomycin therapy after surgery, the patient developed a fever again, this time accompanied by neck stiffness, nausea, an acute headache, and widespread hoarseness.

Her vital signs were as follows: blood pressure—120/70 mmHg; respiratory rate (RR)—32 breaths per minute; body temperature—39 °C; and heart rate (HR)—105 bpm. The patient was no longer in the post-induction aplastic phase, with 11,050 white blood cells (neutrophils 8.790/mm^3^, 79.6%), hemoglobin at 10.1 g/dL, 260,000 platelets/mm^3^, CRP levels at 6.2 mg/L, and procalcitonin (PCT) at 0.3 µcg/L. The lactate levels in the arterial blood gas analysis were 1.6 millimoles/liter. The CSF was cloudy, with 110/mm3 white blood cells, hyperprotidorrachia (554 mg/L), and hypoglycorrhachia (29.9 mg/dL). The Pandy reaction was positive, and culture analysis revealed vancomycin-resistant *E. faecium* growth. On a T2 magnetic resonance (T2MR) assay of her blood, vancomycin-resistant *E. faecium* was also discovered, and the patient tested positive for rectal colonization by KPC *K. pneumoniae*. Simultaneously, a chest CT scan revealed pneumonia and minimal bilateral pleural effusion. Therefore, daptomycin was withdrawn, meropenem was maintained, and linezolid was started, with an initial improvement. In addition, the ventricular drain and all the intravenous (IV) lines were changed. After a few days, the patient’s clinical condition deteriorated further, with the appearance of fever (40 °C), sensorial clouding, severe hypotension (blood pressure 80/50 mmHg), HR of 127 bpm, Sat at 95% aa, RR of 36 breaths/min, lactates on the arterial blood gas analysis (5 millimoles/L), decreased platelets, and an increase in white blood cells (WBC: 18.250/mm^3^; neutrophils: 94.7%) and PCT (>100 µcg/L). The CSF’s chemical–physical and culture examinations were normal. A supportive therapy of fluids and norepinephrine was started, as well as a T2MR assay, and blood cultures were drawn again, yielding carbapenem-resistant, KPC-producing *K. pneumoniae*. Based on the antibiogram, the therapy was adjusted by substituting meropenem with ceftazidime/avibactam (minimum inhibitory concentration (MIC): 4/4 mg/L), and linezolid was continued. However, following the first dose of ceftazidime/avibactam, the patient began to complain of general malaise and excessive sweating, and a diffuse rash developed. As the result, the treatment was adjusted further, and treatment with meropenem/vaborbactam (MIC ≤ 2/8 mg/L) was begun, along with fosfomycin and linezolid, resulting in clinical improvement, normalization of the inflammatory indices, and negative blood cultures. Because the patient still needed a shunt, a ventriculoperitoneal shunt was provided, and the patient was discharged at the end of January. Serial examinations of her bone marrow revealed a significant response to the induction regimen and a stable hematological picture, and the patient underwent a consolidation cycle at the end of February. During the post-consolidation aplasia phase, the patient developed fever (40 °C) in the absence of abdominal pain and symptoms of meningeal inflammation, with a CRP level at 327 mg/L and PCT at 6.36 µcg/L. Her blood cultures returned positive results for KPC *K. pneumoniae* again. Meropenem/vaborbactam treatment was begun and maintained for a total of 10 days, with negative blood cultures after 72 h. Six days after discharge, the patient presented at the emergency department with a high fever, abdominal pain, an abdomen that was tender, and an increase in the inflammatory indices, and a CT scan revealed endoabdominal effusion with an organized appearance and collections in the mesogastric region and the pelvis. As a result, the patient underwent drainage and culture. Post-procedural blood cultures were also drawn. Both were negative. Based on the patient’s microbiological history, meropenem/vaborbactam and linezolid were reintroduced into her therapy and continued for 14 days; however, despite the documented radiological improvement, the patient developed an intense headache, vomiting, rigor nucalis, fever, and a progressive deterioration of neurological objectivity. CSF testing was conducted after the ventriculoperitoneal drain was replaced with an external ventricular drain, and it revealed 2070 white blood cells/mm3—predominantly neutrophils and hyperprotidorrachia (2.190 mg/L). KPC *K. pneumoniae* and vancomycin-resistant *E. faecium* grew in the CSF culture, and *Candida albicans* grew from the tip of the peritoneal catheter. Rifampicin and liposomal B amphotericin were added to the meropenem/vaborbactam and linezolid and continued for 14 days until complete clinical recovery, liquor cellularity reduction, and the absence of abdominal collections on the control CT scan were attained. A new ventriculoperitoneal shunt with a Codman Certas valve calibrated to four was installed.

Ten days later, following a recurrence of fever and abdominal pain symptoms, as well as a sudden deterioration of neurological objectivity, the patient underwent another CT scan of the abdomen with contrast medium, which revealed the presence of an extensive collection (anteroposterior diameter 13.7 cm x transverse diameter 7.4 cm x lateral diameter 10.8 cm) at the distal extremity of the ventriculoperitoneal shunt. At the source control, KPC *K. pneumoniae* was found in the drained collection and in the CSF. Contrast-enhanced brain resonance imaging revealed a picture of tetraventricular hydrocephalus and periventricular hyperintensity in the fluid-attenuated inversion recovery (FLAIR), which was attributed to transependymal exudation phenomena, as well as the flattening of the adjacent periencephalic sulci, indicating ventriculitis [Figure 1]. Again, meropenem/vaborbactam therapy was started and continued for 21 days, and after multidisciplinary evaluation, the placement of ventriculoatrial derivation with the Codman Hakim system (calibrated to 110 mmH_2_O) was decided upon. Afterwards, no further infectious episodes were recorded. Figure 2 displays a timeline describing the clinical history. The patient experienced failure of the first-line therapy for acute leukemia, and a second-line therapy with azacitidine and venetoclax was started. In November 2023, she successfully received an allogeneic bone marrow transplant. The patient is doing well and is being followed up at our hospital’s Hematology Unit.

## 3. Discussion

Infectious complications in hematological patients are frequently an emergency, and death is associated with a delay in diagnosis and the administration of effective antibiotic therapy. However, the need to begin effective antibiotic therapy at an early stage necessitates stewardship and, if feasible, rapid de-escalation. We present the case of a young woman with acute leukemia who had KPC *K. pneumoniae* and vancomycin-resistant *E. faecium* ventriculitis and was successfully treated with meropenem/vaborbactam, linezolid, and rifampicin.

Ventriculitis is an inflammation of the cerebral ventricle’s ependymal lining. There are currently no definite diagnostic criteria for ventriculitis, and this nosological entity is particularly important in patients with EVDs or intraventricular shunts. Catheter-related ventriculitis is linked to considerable morbidity and death, particularly when Gram-negative organisms are involved (up to 58% in certain studies), and the growth of biofilms on the devices shields germs from the host immune response and antimicrobial treatment [2]. Furthermore, the incidence of MDR Gram-negative bacteria as the cause of post-neurosurgical ventriculitis and meningitis has increased in numerous centers [18,19,20]. In our patient, meningitis and ventriculitis were caused by both KPC *K. pneumoniae* and *E. faecium*.

It should be noted that in 2022, 60.8% of the *Klebsiella pneumoniae* isolates at the University Hospital of Palermo were carbapenem-resistant [21].

Enterococcal meningitis is uncommon and linked with a high death rate [22,23,24]. The risk factors for developing post-neurosurgical infections vary depending on the surgical procedure used and include multiple EVDs, longer drainage duration, higher frequency of CSF sampling, prior brain surgery, lower Glasgow coma scale (GCS), and insertion site dehiscence [4]. Infections associated with intracranial implants may be caused by the contamination and/or colonization of the implanted devices [22].

Our patient had multiple risk factors for ventriculitis and infections due to MDR Gram-positive and Gram-negative bacteria, including an underlying immunosuppression related to acute leukemia, a previous neurosurgical intervention with shunt placement, and a lengthy hospitalization. Different mechanisms, most likely retrograde infection, may have contributed to the shunt infection, while hematogenous spread and contamination during surgery cannot be ruled out.

Gram-negative bacillary meningitis is difficult to treat because of its slow progression and tendency to recur [2], particularly in immunocompromised patients. Only a few cases of vancomycin-resistant enterococcal meningitis/ventriculitis have been described in the literature. There are limited therapeutic options for device-associated vancomycin-resistant CNS infections, and the optimal treatment has yet to be discovered [22,23,24,25,26,27]. An antibiotic’s entrance into the CSF is determined by the drug’s physicochemical qualities, alongside host variables such as the patient’s age, CSF flow and volume, plasma albumin, and polymorphisms in genes that encode transport proteins [22].

To the best of our knowledge, there are no ongoing trials on nosocomial ventriculitis and the use of meropenem vaborbactam in central nervous system infections, and this is the fourth case report in which MVB was successfully utilized to treat a post-neurosurgical infection and the second in which MVB was used alone rather than in systemic or intrathecal combination therapy. Choi S. et al. recently reported the case of a 69-year-old patient with multiple co-morbidities who underwent cranioplasty complicated by the formation of a subdural and epidural abscess collection due to KPC *K. pneumoniae*, which was treated with MVB for 25 days (followed by 12 days of CAZ/AVI to comply with the formulary system of the rehabilitation center) with microbiological eradication and clinical stabilization, allowing the patient to be discharged [10]. The choice of MVB in our case was driven by the patient’s adverse response with malaise and the emergence of a rash after the first infusion of CAZ/AVI, whereas the decision in the case described by Choi S. et al. [10] was motivated by the MICs (respectively, 0.064 mg/L and 2 mg/L for MVB and CAZ/AVI). However, the meropenem and vaborbactam levels in the CSF were not available in either of the two patients, and there are also just a few case reports in the literature where CAZ/AVI has been used to treat CNS infections [11,12,13,14,15,16,17]. Volpicelli L. and colleagues recently described a CNS infection caused by CAZ/AVI-resistant, KPC-producing *K. pneumoniae* that was successfully treated with MVB, intravenous fosfomycin, and dosed meropenem and vaborbactam concentrations in the plasma and cerebrospinal fluid [13]. Vaborbactam was shown to permeate the CSF fluid effectively in all the measurements, while meropenem was undetectable. However, the low CNS penetration of meropenem and substantial inter-individual variability have already been observed in the literature [28,29,30,31]. Unfortunately, for our patient, it was not possible to conduct therapeutic drug monitoring (TDM) due to laboratory limitations throughout the hospital stay.

Rezzonico L.F. et al. recently described a case of KPC *K. pneumoniae* post-surgical meningitis treated with MVB, systemic gentamicin, and intrathecal gentamicin [15].

There are no other data on vaborbactam CSF exposure in animal models or people, but its poor protein binding and small molecular weight imply the possibility of CSF exposure [31,32]. MVB was also utilized to treat central nervous system infections in two more cases. The first patient, with a poor outcome, had post-surgical meningitis caused by carbapenem-resistant KPC *K. pneumoniae*, while the other involved a patient with a KPC *K. pneumoniae* meningitis caused by chronic otitis media. In the latter, the MVB treatment presumably lasted for a few days [11,12].

Table 1 summarizes the characteristics of the patients with central nervous system (CNS) infection due to carbapenem-resistant *K. pneumoniae* who were treated with MVB.

Enterococci live in both the human gastrointestinal tract and the environment.

Systemic antibiotic treatment has been found to modify the gut flora in favor of enterococci due to their intrinsic resistance to a variety of antimicrobials.

Broad-spectrum antibiotics with great action against anaerobes but low activity against enterococci may enhance the colonization of the gastrointestinal tract with VRE [23,24,25,26].

Prior to the development of bacteremia and the discovery of vancomycin-resistant *E. faecium* in the CSF, our patient had received therapy with piperacillin/tazobactam, meropenem, and levofloxacin, which may have impacted the development of VRE. Due to the limited CNS penetration and reports of treatment failure, monotherapy is not recommended for VRE meningitis [22,23,24,25,26]. Because of its better penetration into the CNS compared to other treatments, linezolid is frequently regarded as the treatment of choice for individuals with VRE meningitis, if they are susceptible. In our patient, *E. faecium* had a daptomycin MIC of 4 mg/L. For daptomycin, the susceptible dose-dependent breakpoint of ≤4 mg/L, as established by the Clinical and Laboratory Standards Institute, requires a high-dose daptomycin regimen of 12 mg/kg/day for serious *E. faecium* infection. Moreover, for *E. faecium*, there are no EUCAST breakpoints for daptomycin, and this, even at a high dosage, might fail. On the other hand, because of its biofilm action, rifampicin is particularly useful in the treatment of device-associated infections [3,25,26,33,34]. Despite worries regarding potential myelosuppression from linezolid [35], we decided to treat our patient with both linezolid and rifampicin.

Although we did not utilize intrathecal therapy in our patient, this should also be considered for patients with MDR CNS infection and for those who are refractory to systemic therapy [2,3,7].

For device-associated CNS infection, it is suggested that all the components of the infected shunt or EVD be removed, followed by antimicrobial therapy [1,2,3,4]. Immunocompromised individuals require aggressive therapy; hence, our patient had the infected device removed multiple times. Although it is uncertain how long the therapy should last, we continued each time for at least 14 days, and, as advised in the most recent guidelines [1], the ventriculoperitoneal and ventriculoatrial leads were reimplanted following the negative result of the CSF culture test.

The decision to re-implant the device should be based on the patient, microorganism, severity of infection, and CSF findings [1,2]. Moreover, ventriculitis patients must be closely monitored for an extended period because the ventricles and choroid plexus can act as a reservoir of infection, even if a lumbar puncture yields sterile culture results [14].

Finally, this case report emphasizes the importance of further inquiry into the treatment of device-associated ventriculitis and other CNS infections caused by MDR microorganisms. Despite the immunosuppression and the presence of a device, employing meropenem vaborbactam, rifampicin, and linezolid in our patient yielded a favorable outcome, allowing for a successful allogeneic transplant without further infection episodes to date (she continues to be monitored at the Hematology Unit).

## 4. Conclusions

The management of complex infections caused by MDR bacteria, such as device-associated ventriculitis, requires an integrated approach that includes control of the infectious focus and the choice of an adequate antibiotic therapy for an appropriate duration. MVB presents a promising opportunity against ventriculitis caused by KPC *K. pneumoniae*, while the combination of linezolid and rifampicin may be a better choice against ventriculitis caused by vancomycin-resistant *E. faecium*.

## Figures and Tables

**Figure 1 antibiotics-13-00432-f001:**
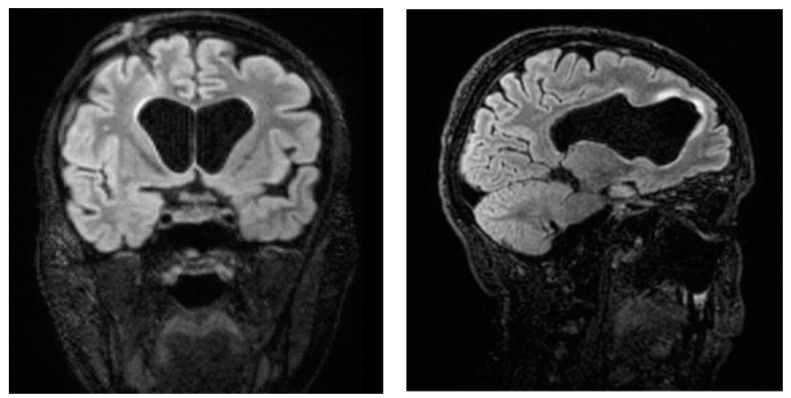
Contrast-enhanced brain resonance imaging revealing a picture of ventriculitis.

**Figure 2 antibiotics-13-00432-f002:**
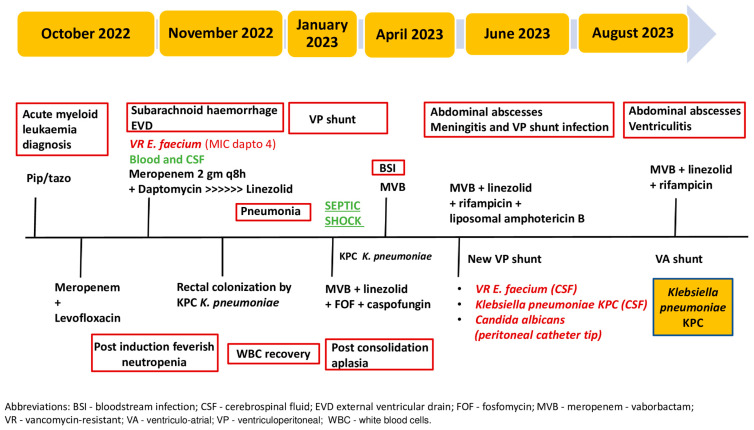
Timeline describing the clinical history.

**Table 1 antibiotics-13-00432-t001:** Characteristics of patients with central nervous system (CNS) infection due to carbapenem-resistant *Klebsiella pneumoniae* treated with meropenem/vaborbactam. Abbreviations: EVD—external ventricular drain; FOF—fosfomycin; MVB—meropenem/vaborbactam; CAZ/AVI—ceftazidime/avibactam; MICs—minimum inhibitory concentrations.

Authors [Reference]	Patient Sex, Age	Underlying Condition	Treatment (Days of Treatment)	Outcome
Rezzonico LF 2024 [15]	Male, 59 years	Peri-mesencephalic sub-arachnoid hemorrhage, post-neurosurgical meningitis, EVD	MVB (21 days) + systemic gentamicin (5 days) + intrathecal gentamicin (from day 5 of therapy till day 10)	Discharged to a neuro rehabilitation center without relapse (four months after discharge)
Volpicelli L 2024 [13]	Male, 27 years	Subarachnoid hemorrhage, post-neurosurgical meningitis, EVD	MVB + FOF, no intraventricular therapy (42 days)	Five months after hospital admission the patient died because of *Pseudomonas aeruginosa* pneumonia
Choi S 2022 [10]	Male, 69 years	Subdural hematoma, cranioplasty, epidural and subdural fluid collection and subgaleal collection, no EVD	MVB, no intraventricular therapy (25 days)	Discharged from hospital with sequelae
Anwar S 2020 [12]	Female, 68 years	Acute left mastoiditis and otitis media, abscess formation in the left petrous apex, meningitis	MVB + systemic polymyxin (presumably for a few days)	Resolution of symptoms and radiological improvement
Yasmin M 2020 [11]	Male, 38 years	Head trauma with a left temporal bone fracture and intra-cranial bleeding, intrathecal pump placement, ventriculitis and left temporal fluid collection	MVB (10 days) + ciprofloxacin (7 days) replaced by CAZ/AVI + amikacin due to more favorable MICs	Gradual clinical improvement

## Data Availability

The data presented in this study are available on request from the corresponding author due to privacy and ethical reasons.

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
