# Peer review of "A Difficult Case of Ventriculitis in a 40-Year-Old Woman with Acute Myeloid Leukemia"

_antibiotics, 2024, doi:10.3390/antibiotics13050432_

Round 1

Reviewer 1 Report

Comments and Suggestions for Authors

In the present manuscript, the authors reported a case of ventriculitis and bacteremia caused by carbapenem-resistant KPC producing Klebsiella pneumoniae and vancomycin-resistant Enterococcus faecium. The case is well written,  presented and discussed, with well-founded conclusions that highlight the importance of an integrated approach that includes control of the infectious focus and the choice of an appropriate antibiotic therapy for an adequate duration.

Some minor comments:

Line 50: “K. pneumoniae” change to “Klebsiella pneumoniae

Lines 56: “Vancomycin-resistant enterococci (VRE)”; line 59: “vancomycin-resistant enterococcus (VRE). Please be consistent.

Line 74: please define “CT”.

Line 83: “Enterococcus faecium” should be in italics.

Line 152: please define “FLAIR”.

Figure 2: Please consider changing the title to “Timeline describing the clinical history” and review the use of italics in “colonization by KPC K. pneumoniae”.

Line 197: please define “GCS”.

Line 225: “meropenem-vaborbactam” change to “MVB”.

Lines 225, 227 and 229: “ceftazidime-avibactam” change to “CAZ-AVI”.

Line 281: “central nervous system infections” change to “CNS infections”;  “mul-tidrug-resistant” change to “MDR”.

Line 287: “Klebsiella pneumoniae” should be in italics.

Reviewer 2 Report

Comments and Suggestions for Authors

The authors present a very illustrative clinical case of severe meningitis and ventriculitis infection by multidrug-resistant microorganisms such as carbapenem-resistant KPC producing Klebsiella pneumoniae and vancomycin-resistant Enterococcus faecium in a very fragile patient with acute leukemia. They review the different treatment options, assessing important aspects such as the diffusion of the different antimicrobials to the central nervous system. The authors also make a review of the literature of similar cases, given the little experience in this profile of microorganisms, in this type of patients and with special localization of the infection such as the central nervous system. The clinical case presented can be very useful for clinicians with less experience in the treatment of this type of infections and for this reason, in my opinion, it meets the criteria for publication.

Author Response

See the attaced file

Reviewer 3 Report

Comments and Suggestions for Authors

The submitted manuscript for a case report of a difficult case of ventriculitis in a young woman with acute myeloid leukaemia, the authors should consider the followings:

Basic lab tests results and background antibiotics resistance rate of the hospital setting or the neighbourhood community of the patient should be available to this article.

Timeline of the patient journey showed a good presentation in Figure 2.

The authors should discuss the limitations of this case report, of the missing results or tests that could have put forward to better manage the case.

The authors should discuss the current situation of the patient.

Prevalence of AML and the ventriculitis of the region should be provided in this article.

A brief literature search of the approach adopted by others and their results may be summarized. Further search on the relevant clinical studies by this approach in clinical trial registries may be searched and summarised to enrich the disscusion.

A 40-year-old maybe used in the manuscript title instead.

In Table 1, please add the info of the ethnicity and year of the case, if available.

Comments on the Quality of English Language

Quality of English Language can be improved by English editing professional.
